# Small Molecule Inhibitors as Therapeutic Agents Targeting Oncogenic Fusion Proteins: Current Status and Clinical

**DOI:** 10.3390/molecules28124672

**Published:** 2023-06-09

**Authors:** Yichao Kong, Caihong Jiang, Guifeng Wei, Kai Sun, Ruijie Wang, Ting Qiu

**Affiliations:** 1School of Pharmacy, Hangzhou Normal University, Hangzhou 311121, China; 2022011010022@stu.hznu.edu.cn (Y.K.); 2022112025023@stu.hznu.edu.cn (C.J.); weiguifeng@stu.hznu.edu.cn (G.W.); 2021112012278@stu.hznu.edu.cn (K.S.); 2021112012297@stu.hznu.edu.cn (R.W.); 2Key Laboratory of Elemene Class Anti-Cancer Chinese Medicines, Engineering Laboratory of Development and Application of Traditional Chinese Medicines, Collaborative Innovation Center of Traditional Chinese Medicines of Zhejiang Province, Hangzhou Normal University, Hangzhou 311121, China

**Keywords:** fusion proteins, inhibitor, therapeutic, cancer

## Abstract

Oncogenic fusion proteins, arising from chromosomal rearrangements, have emerged as prominent drivers of tumorigenesis and crucial therapeutic targets in cancer research. In recent years, the potential of small molecular inhibitors in selectively targeting fusion proteins has exhibited significant prospects, offering a novel approach to combat malignancies harboring these aberrant molecular entities. This review provides a comprehensive overview of the current state of small molecular inhibitors as therapeutic agents for oncogenic fusion proteins. We discuss the rationale for targeting fusion proteins, elucidate the mechanism of action of inhibitors, assess the challenges associated with their utilization, and provide a summary of the clinical progress achieved thus far. The objective is to provide the medicinal community with current and pertinent information and to expedite the drug discovery programs in this area.

## 1. Introduction

Genomic instability is a prominent factor in driving tumorigenesis [1]. Specifically, gene fusions represent significant genomic events in the progression of cancer [2]. Gene fusions result from the physical juxtaposition of two or more previously separate genes, generating a novel chimeric gene that comprises the head of one gene and the tail of another. These chimeric genes can be generated through several possible processes, including chromosomal inversions, tandem duplications, interstitial deletions, or translocation events [3]. Gene fusions often encode fusion proteins that possess unique functional properties distinct from the original proteins, thereby leading to alterations in cellular functions and promoting tumorigenesis. Gene fusions are frequently identified as primary tumor predisposing events in leukemias, lymphomas, solid malignancies, and benign tumors [4]. Various fusion proteins have emerged as critical biomarkers for clinical diagnosis and therapeutic targets [5]. Therefore, gene fusions have garnered substantial attention within the realm of cancer research.

The discovery of the BCR–ABL fusion protein in patients with chronic myeloid leukemia (CML) marked the inception of a new era in cancer research [6]. Subsequently, numerous gene fusions have been identified in various cancer types [7], with advancements in detection methods. The application of RNA sequencing technology has revealed a multitude of unique fusion events, surpassing a count of 2200, with 120 of them exhibiting high frequency. Among these, a substantial proportion encodes protein kinases, which are known to play crucial roles in multiple signaling pathways, with relatively high expression levels and kinase activity [8]. Fusion proteins have been recognized as pivotal drivers of cancer initiation and progression in a variety of malignancies. For instance, the BCR–ABL fusion protein has been shown to induce the transformation of benign cells into malignant ones [9]. In fibrolamellar hepatocellular carcinoma (FL-HCC), the DNAJB1–PRKACA fusion drives tumorigenesis by hyperactivating the Wnt signaling pathway [10]. RET/PTC have been observed in papillary carcinomas and shown to activate the RAS-MAPK pathway in a ligand-independent way [11]. Furthermore, MAN2A1–FER has been identified in various tumor types and has been demonstrated to enhance the FER kinase activation [12]. An increasing number of studies have illustrated the pivotal role of kinase fusion proteins in the pathogenesis of cancer by aberrantly activating downstream signaling pathways. Notably, a mounting body of research has also highlighted the significance of non-kinase fusion proteins, particularly transcription factor-related non-kinase fusion proteins, in cancer development. For instance, PML–RARα and FUS–DDIT3 have been considered as the key drivers of acute myeloid leukemia (AML) and mucinous liposarcoma (MLS) [13,14]. Therefore, comprehending the mechanistic and evolutionary processes underlying fusion proteins is crucial for understanding their role in driving carcinogenesis.

The remarkable specificity exhibited by fusion genes towards cancer cells has propelled their emergence as diagnostic markers and promising targets for cancer therapies. By targeting fusion proteins, the potential arises to mitigate the off-target effects commonly associated with traditional cancer treatments, such as chemotherapy and radiation therapy, which frequently lead to many side effects and adversely impact the patients’ quality of life. Consequently, the development of selective inhibitors for fusion proteins has become a critical goal in the realm of related-cancers therapies. Targeted therapies, particularly tyrosine kinase inhibitors (TKIs), have demonstrated remarkably efficacy in treating specific cancer types, including the first-, second- and third-generation TKIs. For example, Imatinib, a drug designed to target BCR–ABL, has exhibited effectiveness in the treatment of CML [15]. Recently approved drugs such as Larotrectinib and Entrectinib are capable of targeting fusion proteins harboring neurotrophic RTK (NRTK) genes [16]. Crizotinib has demonstrated efficient inhibition of ALK fusions [17]. Additionally, all-trans-retinoic acid (ATRA) and arsenic trioxide (ATO) can be employed to treat or alleviate the blockage of granulocyte differentiation caused by the RARA fusion protein [18]. Despite the success achieved by inhibitors in the treatment of fusion-driven tumors, resistance often emerges as a result of secondary mutations [19]. Remarkably, the therapeutic strategy for cancers has undergone a paradigm shift with the advent of molecular-targeted therapy for oncogenes-driven cancers.

As mentioned above, fusion proteins arising from fusion genes are important targets for cancer therapy. Therefore, in this article, we summarize the techniques for the detection of fusion genes. Furthermore, we present a comprehensive summary of fusion gene occurrences across various cancer types, along with an analysis of the relative frequencies of distinct fusion forms, utilizing the TCGA fusion gene database as our primary data source [20]. Moreover, a compilation of fusion protein inhibitors and an analysis of their corresponding clinical investigations are presented, drawing insights from an extensive search of the DrugBank database and the ClinicalTrials.gov results database.

## 2. Fusion Genes in Diseases and the Detection of Fusion Genes

At the genomic level, the process of chromosome breakage and subsequent re-splicing gives rise to gene fusions, which are novel genes formed through this mechanism [21]. Unlike other recurrent oncogenic mutations, gene fusions are often regarded as pathognomonic factors for malignancy, and many of them are strongly associated with specific cancer types, such as prostate cancer [22,23], lung cancer [24,25], breast cancer, and so on (Figure 1).

Thus, the detection of gene fusions is essential for the identification of cancer and the development of targeted therapeutic approaches. These gene fusions generate novel chimeric proteins that can drive oncogenic transformation. The identification of gene fusions in cancer patients can provide valuable insight for personalized treatment strategies that target the aberrant fusion proteins or downstream signaling pathways, ultimately leading to improved therapeutic outcomes. Currently, a variety of techniques are wildly employed in clinical practice, include cell morphology, exon array analysis (EAA), immunohistochemistry (IHC), fluorescence in situ hybridization (FISH), reverse transcription polymerase chain reaction (RT-PCR), droplet digital PCR (DD PCR), and next-generation sequencing (NGS) (Figure 2).

NTRK, ALK, NTRK1, BRAF, and RET fusions and many other gene fusions have previously been successfully detected by cytomorphological methods [26,27]. However, it may not always be sensitive enough to detect low-level fusion transcriptions or may not be able to differentiate between different fusion variants. The convenience and speed offered by IHC, the specificity provided by FISH, and the high detection sensitivity of RT-PCR have positioned them as widely utilized detection methods. These techniques have made great contributions to the identification of gene fusions, including ALK, NTRK, and RET. While IHC stands out for its cost-effectiveness and rapidity compared to other available assays, caution is warranted due to its tendency to yield false-positive results. Consequently, validation in combination with complementary techniques is often necessary to ensure reliable outcomes [28]. The emergence of NGS technology has facilitated targeted therapy for tumors by enabling the detection not only of fusions but also polygenes and fusion mutants [29]. In addition to the classical detection techniques, a large number of new assays are emerging, including targeted RNA sequencing (RNAseq) and full-length single-cell sequencing (scRNA-seq). Compared to FISH and RT-PCR methods, targeted RNAseq improves diagnostic yield by evaluating multiple genes simultaneously in a single assay, thereby increasing sequencing coverage and reducing diagnostic time. Additionally, it can also identify new gene fusions or complex structural rearrangements. However, the broader scope of detection in targeted RNAseq may lead to an increased likelihood of false-positive results [30]. The scRNA-seq [31], employing the detection algorithm scFusion, presents the opportunity to detect fusion genes at the single-cell level, thereby enhancing the detection power through co-analysis of relevant cells. However, the ability to detect rare fusions in highly heterogeneous tumor samples is limited. Furthermore, scRNAseq data often exhibit high levels of noise and contain various technical artifacts, which can also contribute to false-positive results [31].

## 3. The Development of Inhibitors Targeted Kinase Fusion Proteins

### 3.1. ABL-Fusion Inhibitors

Abelson Murine Leukemia Viral Oncogene Homolo (ABL), a non-receptor tyrosine kinase, participated in many signaling pathways in both the cytoplasm and the nucleus, regulating critical cellular processes such as cell growth, survival, invasion, adhesion, and migration [32]. The most common fusion partner with ABL is the BCR gene, which gives rise to various forms of leukaemia in humans (Figure 3A). The coiled-coil domain of the BCR facilitates dimerization and constitutive autophosphorylation of the ABL tyrosine kinase domain, resulting in uncontrolled tyrosine kinase activity. BCR–ABL has been identified as an anti-apoptotic gene, leading to abnormal cell proliferation and disruption of normal cell regulation. This fusion has been validated as a carcinogenic alteration [33,34]. BCR–ABL fusions are predominantly detected in CML, accounting for approximately 95% of cases. It has also been detected in patients with adult B-cell acute lymphoblastic leukemia (B-ALL) and mixed phenotype acute leukemia (MPAA), with frequencies of approximately 25% and 30%, respectively. In contrast, it is rare in patients with AML, with less than 1% of cases being observed. Additionally, occasional cases of this oncogenic fusion have been reported in patients with lymphoma and myeloma [35]. The primary treatment for BCR–ABL-positive leukemias is the use of TKIs such as Imatinib, Nilotinib, Dasatinib, and Bosutinib (Figure 3B,C) [34].

#### 3.1.1. Imatinib

Imatinib is a 2-phenylaminopyrimidine derivative neoplastic agent that has been approved by the FDA for the treatment of CML by inhibiting BCR–ABL tyrosine kinase [36]. Its mechanism of action involves competing the ATP binding sites of BCR–ABL, thus preventing downstream phosphorylation of target protein. Additionally, Imatinib is an inhibitor of platelet-derived growth factor (PDGF) and stem cell factor (SCF). Despite its effectiveness in treating CML, Imatinib has been reported to cause acquired drug resistance. The development of resistance to Imatinib in CML is a multifactorial process, including point mutations in the structural domain of BCR–ABL kinase, amplification of the BCR–ABL gene, and the presence of leukemic stem cells that can produce Imatinib-insensitive active mutants [36,37]. It is noteworthy that mutations in the structural domain of the BCR kinase (S/T kinase), such as T315I, E255K, Y253F, and M351T, represent a primary cause of drug resistance [36]. During the course of treatment, Imatinib exhibits a relatively low incidence of severe adverse effects, although the emergence of drug resistance remains a possibility. The most frequently observed side effects include muscle cramps, diarrhea, nausea, rash, and bone marrow suppression [38]. Additionally, Imatinib has been associated with the occurrence of cardiovascular complications, particularly heart failure [39].

#### 3.1.2. Dasatinib

Dasatinib is a new and potent multitargeted kinase inhibitor that has been approved as a second-generation TKI for the treatment of CML and ALL [40], which have not responded to Imatinib therapy. In contrast to Imatinib, Dasatinib represses both the active and inactive conformations of the ABL kinase domain. As a result of this unique characteristic, Dasatinib has emerged as a viable therapeutic alternative for patients who exhibit resistance to Imatinib. Currently, Dasatinib is being used as a treatment option for such patients [41]. Similarly, the therapeutic effect of Dasatinib is achieved through the ATP-competitive pathway [42]. In addition, the capability to cross the blood–brain barrier makes it a promising treatment for central nervous system (CNS) leukemia. Frequent occurrence of T315I and F317L mutations has been observed in patients who exhibit resistance to Dasatinib [40,43]. In patients with advanced CML treated with Dasatinib, Pleural effusions have been reported, which can be effectively managed through the use of diuretics and steroids [44]. The incidences of heart failure and cardiomyopathy associated with Dasatinib are comparable to those observed with Imatinib [39]. In recent years, Dasatinib has been found to induce autophagy in BCR–ABL-positive leukemia cells, indicating that autophagy may be a potential direction for the development of novel drugs to treat BCR–ABL-positive leukemia [44].

#### 3.1.3. Nilotinib

Nilotinib is a transduction inhibitor that targets several proteins, including BCR–ABL, c-kit, and PDGFR, for the treatment of chronic phase CML. Nilotinib exhibits structural similarities to Imatinib but is 20- to 50-fold more potent against BCR–ABL in phase I clinical trials. Consequently, Nilotinib has been regarded as a relatively safe treatment option that offers significant therapeutic advantages for patients with CML who exhibit resistance to Imatinib [45]. However, instances of pancreatic dysfunction have been reported during Nilotinib treatment, necessitating caution in patients with a history of pancreatitis. Regular monitoring of lipase levels is advised in such cases. Additionally, cases of prolonged QT interval and sudden death have raised concerns associated with Nilotinib administration [46]. In contrast to Dasatinib, Nilotinib is a non-competitive inhibitor of ATP that effectively blocks the autophosphorylation of BCR–ABL at Tyr177, a GRB2 binding site involved in the regulation of multiple signaling pathways implicated in BCR–ABL pathogenesis [47]. Although Nilotinib is effective against most BCR–ABL mutations, it exhibits negligible efficacy against T315I and G250E mutations.

#### 3.1.4. Asciminib

Asciminib is a novel BCR–ABL inhibitor that has been developed to overcome drug resistance and off-target toxicity associated with current inhibitors [48]. Unlike other inhibitors such as Imatinib that bind to the ATP binding site, Asciminib binds to the myristoyl pocket of BCR–ABL [48]. This unique mechanism of action makes Asciminib effective against many Imatinib-resistant BCR–ABL mutants, including the notorious T315I mutation [49]. While some emerging myristate-site mutations remain sensitive to ATP-competitive inhibitors [50], a combination of Asciminib with an ATP-competitive TKI may prove to be a promising strategy to prevent the development of resistance [51]. Notably, side effects associated with Asciminib in clinical settings encompass bone marrow suppression, hypertension, and cardiovascular toxicity [52].

#### 3.1.5. Summary and Prospect

As with the originally identified oncogenic fusion, research on ABL inhibitors is ongoing, with the development of new drugs to effectively treat CML proceeding concurrently [46]. Crystallographic analysis and chemical structural formula analysis of Nilotinib, Dasatinib, and Bosutinib, which were designed based on the Imatinib scaffold, have significantly improved their affinity towards the target protein. However, despite this improvement, all three compounds have exhibited resistance against the T315I mutant. This mutant, known as the gatekeeper mutation, is positioned at the entrance of the ATP-binding site [53,54]. In response to this challenge, Asciminib was developed to specifically bind to the myristoylation pocket of BCR–ABL, thereby avoiding potential off-target toxicity. However, preclinical and clinical data indicate that resistance mutations, such as V468F and I502L, within and around the myristoylation pocket may pose a unique problem for Asciminib [49,55]. Consequently, future drug design efforts should address the issue of overcoming mutants within the myristoylation pocket or explore alternative targets.

Recently developed ATP-competitive TKIs, such as Ponatinib and Vamotinib, have demonstrated the ability to target both the natural and mutant variants of BCR–ABL, including the T315I gatekeeper mutation. Furthermore, Bosutinib has exhibited high activity against BCR–ABL in phase I and II clinical trials, reflected by a >100-fold increase in potency compared with Imatinib [56]. In addition to the continued progress in developing small molecule inhibitors, therapies targeting the BCR–ABL protein degradation pathway have shown efficacy in targeting not only naturally occurring kinase-active BCR–ABL protein but also the mutant forms of BCR–ABL protein [35]. As a novel therapeutic approach, protein degradation strategies show a potential to outperform traditional protein activity inhibition methods.

### 3.2. ALK-Fusion Inhibitors

The identification of anaplastic lymphoma kinase (ALK) was originally accomplished in anaplastic large cell lymphoma (ALCL), which explains its name. The ALK fusions production is associated with the oncogenicity of ALK, which could be related to the altered conformation and phosphorylation of ALK protein [57]. Nevertheless, it is widely acknowledged that the identity of the fusion partner is the key determinant of the oncogenic activity of ALK [58]. The retained kinase domain of ALK, along with the coiled-coil domain present in most fusion partners, enables the dimerization of ALK and its constitutive activation [59]. In addition to the frequently observed fusion types of ALK, such as EML4–ALK, NPM1–ALK, KIF5B–ALK, TFG–ALK, TPM3–ALK, and TPM4–ALK [25,60,61,62], an increasing number of ALK fusion variants have been discovered (Figure 4). ALK fusion proteins are involved in several cellular signaling pathways, such as Ras/extracellular signal-regulated kinase (ERK), phosphatidylinositol 3 kinase (PI3K)/Akt, and Janus protein tyrosine kinase (JAK)/STAT [63] (Figure 5A). Thus, compromised regulation may lead to the development of several diseases. In addition, ALK fusion proteins have been reported in nonelastic ALCL, inflammatory myofibrosarcoma (IMT), diffuse large B-cell lymphoma (DLBCL), and non-small cell lung cancer (NSCLC) [64]. Furthermore, while EML4–ALK fusion is commonly observed in the majority of the 2–7% of NSCLC cases [65], ALK fusions are infrequent in SCLC. Nonetheless, there have been four rare instances of SCLS with ALK fusions reported recently, including ELM4–ALK, PLEKHM2–ALK, and ALK–IR [66,67] (Figure 4). Fortunately, the first-generation drugs, including Crizotinib, a small molecule tyrosine kinase inhibitor initially designed to target cMET, along with the second-generation drugs, such as Ceritinib and Alectinib, and the third-generation agent Lorlatinib, were repurposed effectively for the treatment of cancer with ALK fusion (Figure 5B).

#### 3.2.1. Crizotinib

Crizotinib, the first generation ALK tyrosine kinase inhibitor, was initially designed as a cMET kinase inhibitor. It was subsequently discovered to inhibit ALK-rearranged and ROS-rearranged NSCLC [68]. The use of Crizotinib leads to decreased phosphorylation of ALK fusion, resulting in an inactive protein conformation [69]. In various xenograft models using human-derived tumors or tumor cell lines expressing EML4- or NPM–ALK fusion proteins, Crizotinib inhibited tumor growth, decreased proliferation, increased apoptosis, and dose-dependently decreased phosphorylation. Based on its proven efficacy and safety in clinical trials, Crizotinib was granted approval by the U.S. FDA on 26 August 2011, and received full approval on 20 November 2013 [70]. Despite the significant and long-lasting benefits of Crizotinib, the majority of patients eventually develop resistance, usually after a few years [71]. The main mechanisms of resistance to Crizotinib include secondary resistance mutations in ALK, such as L1196M, C1156Y [72], G1269A [73], L1152R [74], and I1171 [75] (Figure 5B), as well as ALK copy number alterations. Interestingly, Crizotinib has demonstrated sensitivity to a large number of new ALK fusions, including HIP1–ALK [76], BIRC6–ALK [77], and DYSF–ALK [78]. During the use of Crizotinib, elevated transaminase levels and neutropenia may occur [79]. In addition, impairment in renal function and vision has been reported [80,81].

#### 3.2.2. Ceritinib

Ceritinib is a kinase inhibitor for the treatment of patients with ALK-positive metastatic NSCLC who have failed prior Crizotinib therapy due to resistance or intolerance. It was approved by the FDA in April 2014, as a surprisingly high response rate (56%) towards Crizotinib-resistant tumors [82]. Mechanistically, Ceritinib suppresses ALK-mediated phosphorylation of the downstream signaling proteins in an ATP-competitive manner. It has been demonstrated to be effective in treating NSCLC patients who have developed resistance to Crizotinib, specifically, mutations in key “gatekeeper” residues of the enzyme, namely, L1196M, G1269M, and G1269A [83]. These findings suggest that Ceritinib holds promise as a therapeutic option for NSCLC patients with ALK fusion and resistance to Crizotinib. The main mechanisms of resistance to Ceritinib include secondary resistance mutations in ALK, such as G1123S [84], G1202R [85], and F1174C [86]. However, some patients taking Ceritinib may experience gastrointestinal symptoms such as diarrhea and vomiting, as well as elevated transaminase levels, hypophosphatemia, and some hepatotoxicity [79,87,88].

#### 3.2.3. Alectinib

Alectinib, a highly selective inhibitor of ALK, is used specifically in the treatment of NSCLC expressing the EML4–ALK fusion protein [89]. It has been approved in Japan for the treatment of advanced, unresectable, or recurrent NSCLC with ALK rearrangement [90]. Alectinib is frequently used as a second-line treatment after Crizotinib therapy due to drug intolerance or resistance. Furthermore, it has exhibited remarkable effectiveness against several novel ALK fusion proteins, including VKORC1L1–ALK [91], SQSTM1–ALK [92], HIP1–ALK [93] and STRN–ALK [94]. For patients whose disease progression persists despite initial treatment with Crizotinib, converting to Alectinib may be a viable option that could enhance treatment compliance [95]. Patients undergoing Alectinib treatment commonly exhibit elevated levels of creatinine phosphokinase and may experience neutropenia [79]. Notably, the presence of T1151K has been found to increase sensitivity to Alectinib [91]. However, secondary resistance mutations in ALK, including I1171T/N/S [75], G1202R [96], V1180L, and F1174C, may lead to resistance to Alectinib.

#### 3.2.4. Brigatinib

Brigatinib is a tyrosine kinase inhibitor that selectively targets ALK and EGFR and has demonstrated efficacy against nine different Crizotinib-resistant mutants of the EML4–ALK fusion [97]. It was developed by Ariad Pharmaceuticals and received FDA approval in 2017. In BaF3 cells expressing EML4–ALK, and these with gatekeeper mutations (G1269S and L1196M), Brigatinib has been shown to induce tumor regression. Although clinical trial data regarding the use of this drug are currently limited, in EML4–ALK xenograft models in mice, Brigitanib exhibits a dose-dependent inhibition of tumor growth, tumor burden, and prolonged survival [97]. Currently, Brigatinib is primarily used as a treatment option for ALK-positive NSCLC who are intolerant to Crizotinib. Gastrointestinal symptoms, respiratory events, fatigue, cough, and headache are among the commonly reported side effects associated with Brigatinib use [79]. Notably, compared to other ALK-TKIs, Brigatinib has been observed to induce hypertension [81].

#### 3.2.5. Summary and Prospect

At present, inhibitors targeting ALK fusion proteins are categorized into first, second, and third generations. The primary pharmacokinetic challenges encountered with Crizotinib, the first-generation ALK inhibitor, stem from its resistance issues and limited blood–brain barrier permeability [98]. To overcome these limitations, the structurally optimized second-generation inhibitors, Alectinib, Ceritinib, and Brigatinib, were developed. However, the use of second-generation compounds has led to the emergence of drug-resistant mutations in the ALK kinase domain [85]. Lorlatinib, a third-generation inhibitor, represents a small and compact macrocyclic inhibitor that exhibits improved metabolic stability and a low frequency of P-glycoprotein-mediated efflux in in vitro studies. While Lorlatinib demonstrated efficacy, concerns regarding its toxicity have been raised [99]. Despite the availability of the first-, second-, and third-generation ALK-fusion inhibitors and a profound understanding of the resistance mechanisms, drug resistance to ALK-fusion inhibitors remains a major challenge [100]. As illustrated in Figure 4, drug design can be improved by considering the characteristics of the fusion type, particularly in the presence of the coiled-coil domain [101]. Above all, to effectively prevent or overcome drug resistance, a comprehensive understanding of the underlying mechanism is necessary. The development of small molecule drugs, such as Brigatinib, a potent and selective ALK inhibitor, has shown promise in treating metastatic NSCLC patients who have progressed but are intolerant to Crizotinib [102]. Similarly, NVP–TAE684 is also a potent and selective ALK inhibitor that effectively induces apoptosis and cell cycle arrest through rapid and prolonged inhibition of phosphorylation of NVM–ALK and its downstream effectors. Meanwhile, it is sensitive to the gatekeeper mutation L1196M [103]. Therefore, addressing drug resistance issues, reducing drug toxicity, and enhancing drug response rates within the brain constitute essential directions for the advancement of ALK fusion inhibitors.

### 3.3. ROS1 Fusion Inhibitors

ROS1 stands for c-ROS sarcoma carcinogen receptor tyrosine kinase (ROS proto-oncogene 1, receptor tyrosine kinase), which encodes a transmembrane tyrosine kinase receptor called ROS1 protein. However, the functional capabilities of ROS1 remain largely unknown [104]. Fusion events involving ROS1 have been detected in various cancers in adult and pediatric patients. The identified fusion partners include CD74, CCDC6, EZR, FIG, KDELR2, and others [105,106,107]. While fusion events may retain the kinase domains of ROS1, it has been reported that only certain fusion partners, such as FIG, CCDC6, TMP, EZR, and GOPC, contain coiled-coil domains [106]. In addition, it is not clear how several of the fusion partners of ROS1, such as CD74, induce oncogenicity as they lack the ability to induce dimerization in the N-terminal domain [106] (Figure 6). ROS1 fusion induces activation of several typical signaling pathways involved in cell survival and growth, for instance RAS–MAPK, JAK–STAT3, PI3K–AKT–mTOR [108] (Figure 7A). ROS1 fusion proteins account for approximately 1% of NSCLC [109], while in lung adenocarcinoma, the number is about 3.3%. Currently, ROS1 inhibitors are classified into two classes based on their preference for binding to different conformations of the ROS1 kinase domain: type I inhibitors, such as Crizotinib and Entrectinib, preferentially bind to the “DFG-in” conformations of their targets; un contrast, type II ROS1 inhibitors, such as Cabozantinib and Foretinib, have a higher binding affinity for the “DFG-out” conformation [110].

#### 3.3.1. Crizotinib

Crizotinib is a tyrosine kinase receptor inhibitor that has been used for the treatment of ROS1-positive NSCLC tumors. At present, it is the only clinical phase IV inhibitor available for this type of cancer [106]. By competing with ATP, it effectively inhibits the activity of ROS1 fusion with sub-nanomolar cellular potency [111]. Compared to other inhibitors, such as Ceritinib and Alectinib, Crizotinib displayed significantly improved activity against ROS1 kinase [112]. However, it was invalid for G1202R and G2032R [113,114]. Resistance to Crizotinib in cancers may develop due to various mechanism, including secondary mutations in the structural region of the ROS1 kinase or amplification of the ROS1 gene. Additionally, drug resistance may develop due to the activation of other signaling pathways that bypass the ROS1 pathway. To overcome Crizotinib resistance, several second-generation ROS1 inhibitors, including Ceritinib, Roritinib, and Enretinide, are currently undergoing clinical trials [71]. Although Crizotinib has demonstrated effectiveness and good tolerability in ROS1-positive NSCLC patients without major adverse events, the most commonly observed adverse reactions are elevated aspartate/alanine aminotransferases (AST/ALT) and visual impairment [115].

#### 3.3.2. Entrectinib

Entrectinib is a multi-kinase inhibitor that functions as an ATP competitor and has been demonstrated to be effective against ROS1, ALK, and pan-TRK [116]. In 2019, the FDA approved Entrectinib for the treatment of ROS1-positive metastatic NSCLC and NTRK gene fusion-positive solid tumors. An integrated analysis of three-phase 1–2 trials indicated that Entrectinib shows promising results in the treatment of ROS1 fusion-containing cancers and has been found to be 40 times more potent than Crizotinib [117]. The administration of Entrectinib commonly leads to side effects such as congestive heart failure, central nervous system effects, fractures, fatigue, syncope, and taste disturbances [118]. One of the advantages of Entrectinib is the ability to effectively cross the blood–brain barrier, making it a potentially effective treatment option for tumor that have spread to the brain [117]. However, it also encounters the obstacle of drug resistance mutations, such as G2032R, F2004C/I/V, L2086F [119], and L2026M [120].

#### 3.3.3. Summary and Prospect

Regrettably, the number of developed inhibitors for the ROS1 fusion is relatively limited, with only two drugs, Crizotinib and Entrectinib, approved by the FDA (Figure 7B). Crizotinib and Entrectinib belong to type I inhibitors class, which exhibit a preference for binding to the DFG of their respective target. In the active state of ROS1, characterized by the aspartate–phenylalanine–glycine (DFG)-induced conformation, the phenylalanine residue in the DFG motif is positioned within the hydrophobic pocket, immediately preceding the activation loop of the kinase domain [110]. Based on the phase I/II clinical studies, Crizotinib has shown significant therapeutic potential against ROS1-fusion driven cancers [112]. Crizotinib faces challenges primarily associated with resistance mutations and limited brain permeability, while Entrectinib, an indazole derivative. was designed to possess systemic activity and the ability to cross the blood–brain barrier and accumulate within the CNS. Nonetheless, the emergence of acquired resistance to Entrectinib remains inevitable [121]. In addition to the classical inhibitors, numerous drugs undergoing clinical trials have exhibited favorable inhibitory effects. Repotrectinib is a low-molecular-weight macrocyclic TKI with highly selective effects on ROS1 and ALK [122]. Brigatinib, was initially used to treat ALK-positive metastatic NSCLC intolerant to Crizotinib [123], and was later found to have inhibitory activity against ROS1 fusion-positive tumors, making it a next-generation tyrosine kinase [124]. Additionally, besides targeted treatment, programmed death-ligand 1 (PD-L1), an immune-suppressive molecule, needs to be considered for therapeutic decision-making [125]. However, whether the administration of immune checkpoint inhibitor (ICI) therapy is suitable for ROS1 fusion after the development of Crizotinib resistance is uncertain.

### 3.4. NTRK Fusion Inhibitors

NTRK refers to a class of neurotrophic receptor tyrosine kinases that are involved in the regulation of various cellular processes through the activation of intracellular signaling pathways. The NTRK gene family is comprised of three distinct members, namely, NTRK1, NTRK2, and NTRK3, which encode for the TRK family of receptor proteins, including TRKA, TRKB, and TRKC [126]. The fusion types mainly include EML4–NTRK3, TPM3–NTRK1 [127], QKI–NTRK2 [128], etc. NTRK gene fusions results in the production of chimeric oncoproteins such as ETV6–NTRK3 [129] (Figure 8B). These oncoproteins function as oncogenic drivers that promote tumor cell proliferation and survival in vitro. Furthermore, immunohistochemical analysis of TRK expression in NTRK fusion-positive tumors indicates that the subcellular localization of the fusion protein may be determined by the partner [16]. NTRK fusions are frequently observed in rare cancer types including secretory breast cancer and congenital mesodermal nephroma, while they occur less frequently in more common tumors such as breast, lung, and colorectal cancers [130] (Figure 8A). Larotrectinib and Entrectinib have been approved by the FDA and EMA as monotherapy for the treatment of patients with NTRK gene fusion-positive solid tumors (Figure 9B).

#### 3.4.1. Entrectinib

Entrectinib is a multi-kinase inhibitor with ATP-competitive properties that targets TRKA, TRKB, TRKC, ROS1, and ALK. TRK receptors play a role in cell proliferation through downstream signaling of MAPK, PI3K, and PLCγ. By inhibiting these pathways, Entrectinib can suppress cancer cell proliferation, leading to tumor size reduction [131]. The FDA has granted approval for the use of Entrectinib in the treatment of NTRK gene fusion-positive solid tumors in both adult and pediatric patients. The systemic antitumor activity of this compound has also been observed [132]. Without doubt, Entrectinib is associated with certain safety concerns, including congestive heart failure, central nervous system effects, fractures, lung infections, breathing problems, cognitive impairment, fainting, pulmonary embolism, and pleural effusions. The monitoring and management of these side effects are important considerations during treatment with Entrectinib [118]. Clinical studies have shown that Entrectinib is safe and well-tolerated in patients with various tumor types, including NSCLC harboring SQSTM1–NTRK1 [133], metastatic colorectal cancer harboring LMNA–NTRK1 [134], lung cancer harboring SQSTM1–NTRK1 [135], mammary analogue secretory carcinoma harboring ETV6–NTRK3 [136], and glioneuronal tumors harboring BCAN–NTRK1 [137]. However, cases of resistance to Entrectinib have been reported, such as G595R, G623R [138], G667C [138], G623E, V573M, and G595L [138] (Figure 9B).

#### 3.4.2. Larotrectinib

Larotrectinib is a highly selective TRK inhibitor that has been approved for the treatment of NTRK fusion-positive solid tumors in both adults and children [139]. By binding to TRK, Larotrectinib effectively obstructs neurotrophin–TRK interactions and TRK activation, resulting in the induction of cellular apoptosis and the inhibition of tumor cell growth in cases where TRK is overexpressed. Clinical studies have demonstrated that it exhibits potent antitumor activity in patients with NTRK fusion positive cancer, such as those with TPM3–NTRK1 in lung cancer [140] and CTRC–NTRK1 in pancreatic ductal adenocarcinoma [141]. The administration of Larotrectinib has been associated with several common Grade 3 or more serious side effects; these include anemia, elevated alanine aminotransferase or aspartate aminotransferase levels, weight gain, and decreased neutrophil count [142]. Larotrectinib has demonstrated the ability to inhibit the growth of cell lines or xenografts containing TPM3–NTRK1, MPRIP–NTRK1, and ETV6–NTRK3 [140]. Moreover, Larotrectinib exhibited favorable tolerability and demonstrated efficacy across all patients with tumors harboring NTRK gene fusions, thereby representing a promising therapeutic option for such individuals. However, Larotrectinib displayed limited activity in cell lines with point mutations in the TRK kinase domain, such as G667S, G696A, G595R, G623R, F617L [143], and V573M [138].

#### 3.4.3. Summary and Prospect

Larotinib and Entretinib were approved by the FDA for NTRK fusion-positive cancers in 2018 and 2019, respectively. Larotrectinib was developed through structural optimization of a TRKB inhibitor with a novel benzonitrile-substituted imidazopyridine core, guided by structure–activity relationship (SAR) analysis. The resulting Larotrectinib exhibited significant and durable antitumor activity in patients with NTRK fusion cancer [144]. Similarly, Entrectinib was obtained through optimized modification of a 3-amino-5-substituted indazole compound [145]. Both Larotrectinib and Entrectinib belong to the type I kinase inhibitors class and compete with ATP for binding [146]. However, similar to other targeted therapies, the development of acquired resistance limits the efficacy of TRK inhibitors. Mutations in NTRK genes are responsible for drug resistance, including solvent front mutations in TRKA G595R and TRKC G623 in the ATP-binding pocket of the kinase structural domain. To overcome acquired resistance to first-generation TKIs, next-generation TRK inhibitors such as Selitectinib are being developing. In preclinical studies, Merestinib has shown efficacy in tumor models with NTRK fusion and TRKA G667C variants [147]. Additionally, Cabozantinib can target NTRK fusions, but may also exhibit resistance against TRKA L564H, D679G, G595R, and G595L mutations [148]. It is crucial to note that treatment with TRK inhibitors can lead to the overactivation of cancer-related pathways (MAPK pathway), including hotspot mutations, MET amplification, KRAS mutations, and BRAF V600E mutations. This effect may result in treatment failure; thus, combination therapy with drugs that regulate these pathways may represent a new paradigm in the fight against NTRK fusion diseases [148]. Moreover, combination therapy involving NTRK inhibitors and immune checkpoint inhibitors may also offer a promising treatment option.

### 3.5. RET Fusion Inhibitors

Rearranged during transfection (RET), a member of receptor tyrosine kinase family, has been demonstrated to have a crucial role in signal transduction. Perturbations in the expression of RET can lead to tumorigenesis. The most prevalent oncogenic variants include gene fusions and mutations. Notably, studies have highlighted that RET fusions occur with high frequently in populations exposed to ionizing radiation [149]. Various types of RET fusion have been identified, including CDCC6–RET, NCOA4–RET, KIF5B–RET [150], LSM14A–RET [151], TRIM33–RET, ZNF477P–RET [150], etc. (Figure 10B). Typically, the gene fusion encodes a fusion protein that retains the kinase domain, which can induce the overactivation of downstream signaling pathways via partners’ dimerization or oligomerization, culminating in the development of multiple malignancies [152] (Figure 11A). RET fusion proteins have been detected in more than 20 cancer types, with thyroid and lung adenocarcinomas being the most prevalent [150] (Figure 10A). Cabozantinib, Vandetanib, Lenvatinib, Ponatinib, Sunitinib, and Sorafenib have been approved as therapeutic agents for treating RET fusions [153] (Figure 11B).

#### 3.5.1. Lenvatinib

Lenvatinib, an oral tyrosine kinase inhibitor, has been demonstrated to target various receptors, including vascular endothelial growth factor receptor (VEGFR) 1/2/3, fibroblast growth factor receptor (FGFR) 1/2/3, and RET [154]. Current preclinical and clinical studies indicate that it can hinder the autophosphorylation of KIF5B–RET, CCDC6–RET, and NCOA4–RET, which can ultimately prevent the development of tumors harboring RET fusion oncogenes. Its efficacy has been demonstrated in various cancers, including CCDC6–RET-positive human thyroid cancer [155] and lung cancers [154]. Despite these promising findings, targeted therapy for RET fusions has not produced favorable outcomes. Furthermore, it is important to consider the potential adverse effects of Lenvatinib on patient prognosis; hypertension, diarrhea, appetite/weight loss, hand and foot skin reactions, and proteinuria are the most common adverse events associated with Lenvatinib treatment [156]. Over time, resistance to Lenvatinib gradually develops after treatment. Specifically, mutations such as V738A, A807V, F998V, L730V, E732K, and G810A have been identified as potential mechanisms underlying resistance to it [157].

#### 3.5.2. Cabozantinib

Cabozantinib, a multi-kinase inhibitor, has been reported to bind to both the ATP binding site and adjacent hydrophobic/variable sites [152]. Phase II clinical trials in patients with RET fusion-positive NSCLC demonstrated its efficacy [154]. The administration of Cabozantinib is also associated with a range of adverse reactions, including diarrhea, nausea, hypertension, weight loss, muscle tone disorders, hypothyroidism, breathing difficulties, and anemia [158]. However, a comparative analysis has revealed that Cabozantinib exhibits off-target effects and is not specific for RET gene fusions. The development of resistance to Cabozantinib during its use is a potential problem, and specific mutations include L730I, E732K, and V871I [157].

#### 3.5.3. Vandetanib

Vandetanib is a multi-target inhibitor of several proteins, including VEGFR2-3, EGFR, and RET [152]. It has been demonstrated to inhibit the autophosphorylation of these proteins, and then to block the downstream signaling [154]. Inhibition of RET-fused cancers is achieved by competing with ATP [152], leading to a reduction in the colony-forming ability of RET-fused cell lines. Vandetanib has also been found to inhibit the growth of both KIF5B–RET-mediated tumors and multiple-RET-fusion-positive NSCLC [154,159]. Despite its wide usage, the administration of Vandetanib has been associated with serious adverse effects in patients. These adverse effects include QT interval, tachycardia, sudden death, and potential cardiac toxicity [158]. Several drug-resistant mutations have been identified as frequently occurring during use, including G810A, L881V [160], C634W, M918T [152], V804L/M/E [152,161], G810S [157], and S904F [162].

#### 3.5.4. Selpercatinib

Selpercatinib is an orally administered, small-molecule ATP-competitive agent known for its high selectivity [163]. While it is currently being investigated in clinical trials, Selpercatinib obtained accelerated approval from the FDA in 2020 for the treatment of specific RET-driven cancers. It exhibits greater sensitivity in patients with KIF5B–RET type, CCDC6–RET type, and RET V804L/M- and M918T-resistant NSCLC [164]. Its antitumor activity in the CNS is notably prolonged as it is capable of crossing the blood–brain barrier [165]. Notably, the use of Selpercatinib in patients with RET fusion-positive NSCLC who have undergone platinum-based chemotherapy or remained untreated resulted in a significant reduction in tumor growth with minimal toxicity [166]. Selpercatinib has exhibited robust and durable anti-tumor activity in medullary thyroid carcinoma (MTC) cases harboring RET mutations [165]. Importantly, it has been associated with a favorable safety profile [167], although some common adverse reactions have been observed, such as gastrointestinal discomfort, elevated liver enzyme levels, prolonged QT time, abdominal pain, hypertension, and fatigue [168]. However, recent reports have identified mutations in RET G810R/S/C/V that have the potential to result in treatment failures, despite not affecting ATP binding [113,114,162].

#### 3.5.5. Pralsetinib

Pralsetinib is a highly selective, orally administered, small-molecule drug that has demonstrated notable efficacy and tolerability in the treatment of various RET abnormalities, including wild-type and mutations such as V804L, V804M, and M918T, as well as fusions such as KIF5B–RET and CCDC6–RET [169,170]. Similar to Selpercatinib, the phase I/II trial of Pralsetinib is currently ongoing. Nevertheless, Pralsetinib received accelerated approval from the FDA approval in 2020, specifically for the treatment of metastatic NSCLC with RET fusion. It has exhibited considerable therapeutic efficacy in patients with RET fusion-positive metastatic NSCLC, RET mutant MTC, and RET fusion-positive thyroid, with no significant off-target toxicity [154,169]. Moreover, Pralsetinib has been shown to have no genotoxicity both in vitro and in vivo [171], making it a promising RET fusion inhibitor. The most commonly observed treatment-related adverse events of Pralsetinib include elevated aspartate aminotransferase, anemia, constipation, and hypertension [172]. However, caution must be taken when administering it with P-gp (P-glycoprotein) and CYP3A inhibitors, as they may decrease its effectiveness and increase the incidence and severity of adverse reactions [159,169]. Regrettably, it has been found to be insensitive to G810R/S mutation, which may limit its effectiveness in certain patient populations [173].

#### 3.5.6. Summary and Prospect

Multi-kinase inhibitors, such as Cabozantinib, Vandetanib, and Lenvatinib, have demonstrated efficacy in the treatment of RET fusion-positive cancers, but their lack of specificity and susceptibility to drug resistance have been noted. The advent of highly selective targeted agents for RET has improved the treatment of RET fusion-positive patients. The selective inhibitors Pralsetinib and Selpercatinib have been approved by the FDA for the treatment of RET fusion-positive NSCLC [154]. Pralsetinib, a 4-aminopyrimidine derivative, exhibits high affinity towards RET fusion proteins and circumvents interference from RET V804M/L gatekeeper mutations by encompassing the K758M gate–wall residue, thus gaining access to both the front and back clefts [174]. Selpercatinib, a pyrazine derivative, shares a similar high affinity profile to Pralsetinib [162]. Additionally, several RET fusions-targeting inhibitors are being developed and are currently undergoing clinical trials. These include TAS0953 (NCT04683250) for RET fusion positive cells [173], BOS172738 for RET fusion-positive NSCLC [159], RXDX-105 for CCDC6–RET, NCOA4–RET, and PRKAR1A–RET-containing cells [150], and TPX-0046 for advanced solid tumors with RET fusions or mutations [152,159]. In addition, exploring combination therapies may lead to new developments in the treatment of RET fusion mutations.

### 3.6. Others

#### 3.6.1. EGFR Fusion Inhibitors

Epidermal growth factor receptor (EGFR) is a transmembrane glycoprotein and a type I receptor tyrosine kinase that plays a key role in epithelial cell proliferation and signaling. Somatic mutations of EGFR have been found to be associated with various oncogenic processes, including tumor cell proliferation, angiogenesis, tumor invasion, and metastasis [175]. Notably, EGFR fusions have also been observed among these mutations. The most common one is EGFR–RAD51 [175], while other fusions include HMGA2–EGFR [176], KIF5B–EGFR [177], EGFR–SEPT14 [178], EGFR–PURB [179], etc. (Figure 12A). The fusion of EGFR with RAD51/PURB may have the potential to enhance EGFR activity through two mechanisms, namely, asymmetric dimerization promotion and inhibition of receptor switching. The C-terminal tail of EGFR contains multiple autophosphorylation sites that participate in downstream signaling pathways, such as MAPK and PI3K/Akt (Figure 12A). However, in EGFR–RAD51 and EGFR–PURB fusion, the C-terminal domain (CTD) is absent, resulting in a disruption of signal regulation that can lead to pathological consequences [175,179]. For the treatment of EGFR-aberrant disease, including EGFR fusions, EGFR TKIs of the first, second, and third generation, such as Erlotinib, Afatinib, and Osimertinib, have been developed [180] and reported to be effective. The anilino quinazolines analogues, Erlotinib and Gefitinib, belong to the first-generation EGFR TKI. Erlotinib has been shown to inhibit the growth of glioma expressing HMGA2–EGFR and to prolong survival in animal models. It has also demonstrated efficacy against the rare EGFR–SEPT14 fusion protein in patients with colorectal adenocarcinoma [176,181]. Rash and diarrhea are the most common treatment-emergent adverse events, with interstitial pneumonia (ILD) occurring in approximately 1% of patients [182,183]. The second-generation Afatinib, also an anilino quinazoline, has demonstrated a favorable responses in NSCLC patients with KIF5B–EGFR fusion [184]. Common adverse effects associated with Afatinib include diarrhea, rash, paronychia/nail effects, and stomatitis/mucositis [185]. However, the presence of an aniline moiety in the structure of the first- and second-generation inhibitors conflicts with the MET790 side chain, thus rendering the T790M mutation the most common form of resistance [186,187]. In contrast, the third-generation Osimertinib, a mono-anilino pyrimidine compound, has demonstrated sustained CNS activity and the ability to inhibit tumor growth selectively, making it an attractive option for the treatment of EGFR-Fused NSCLC [188]. Its potency against T790M mutants stems from key interactions such as hydrogen bonding with MET793 and covalent bonding with CYS797 [187]. Common adverse effects of Osimertinib include diarrhea, nausea, poor appetite, etc. Patients may also develop leukopenia and lymphocytopenia [187,189,190]. Unfortunately, clinical trial data for the aforementioned compounds are lacking. Lazertinib has been found to be therapeutically similar to Osimertinib [191]. In addition, the use of Cetuximab, an EGFR antibody, has shown promise in inhibiting the proliferation of Ba/F3 cells expressing EGFR–RAD51 [175,179]. Although EGFR fusions are not well understood due to their low incidence and lack of detection technology [188], it is critical to develop more accurate and effective detection methods to improve our understanding of EGFR fusion and its potential in patient treatment.

#### 3.6.2. RAF/BRAF-Fusion Inhibitors

B-rapidly accelerated fibrosarcoma (BRAF) is a member of the serine/threonine protein kinase family that regulates the MAP kinase/ERK signaling pathway, which is involved in various cellular processes, including cell division, differentiation, and secretion. BRAF aberrations, including gene fusions such as KIAA1549–BRAF [192,193], FAM131–BRAF [194], GIT2–BRAF [195], GTF2I–BRAF [196], and LMO7–BRAF [197], have been identified, and the list of novel BRAF fusions continues to grow (Figure 12B). Among these, KIAA1549–BRAF is the most prevalent type of BRAF fusion. RRAF fusions are often found in low-grade gliomas such as dermatocytic astrocytoma (PA), dermatomyosarcoma-like astrocytoma (PMA), and diffuse meningeal glial neuron tumor (DLGNT) [198]. Sorafenib is a potent inhibitor of RAF1 and exhibits effects on BRAF, RET, PDGFRβ, and VEGFR-3. The most common side effects of Sorafenib are fatigue, rash, hair loss, and high blood pressure [199]. In a phase I clinical trial, a combination of Sorafenib and Bevacizumab showed promising results in patients with a KIAA1549–BRAF fusion [200] (Figure 12B). Furthermore, in vitro and in vivo studies have shown that the combination of Trametinib with the PI3K inhibitor BKM120 or the CDK4/6 inhibitor LEE011 produces synergistic antitumor effects in melanoma cells harboring BRAF fusions [201]. The BRAF V600E mutation is a recalcitrant disease commonly observed in melanoma, which has fueled the development of RAF inhibitors targeting this mutation. Second generation inhibitors, such as the azaindole-derivative compounds Vemurafenib and Dabrafenib, and the pyrimidine- derivative compound Encorafenib, have demonstrated significant clinical efficacy and improved survival in patients with BRAFV600E melanoma [202].While Vemurafenib, Dabrafenib, and Encorafenib are selective BRAF inhibitors used in BRAF V600E mutated melanomas [203], limited data exist on their effectiveness against BRAF fusion kinase due to the low occurrence of these alterations in cell lines [201]. Initial research on cell lines suggests that drug combinations may be more successful. Efforts have been made to utilize inhibitors of both BRAF and MEK to overcome drug resistance [204]. Third generation BRAF inhibitors are currently in the development stage and hold promise for the treatment of BRAD fusions and BRAF mutations.

## 4. The Development of Inhibitors Targeted Non-Kinase Fusion Proteins

### 4.1. RARα-Fusion Inhibitors

Retinoic acid receptor α (RARα) is a nuclear hormone receptor that binds to the retinoic acid response element (RARE) and forms a dimer with the retinoid X receptor protein (RXRA), collectively known as the RARα–RXRA complex. The RARα–RXRA complex is essential for promyelocyte differentiation. Acute promyelocytic leukemia (APL) is a subtype of AML. Although, the molecular mechanisms of development and progression remain to be fully understood, the PML–RARα fusion, resulting from the chromosomal translocation t(15;17) (q24;q21), is considered to be the hallmark of APL [205]. PML–RARα fusion has been identified in more than 98% of APL patients, while rare RARα fusions, including NPM–RARα and others, have also been identified. The production of PML–RARα fusion proteins results in the formation of stable homodimers (Figure 13 and Figure 14). Then, these fusion proteins can disrupt normal RARα signaling, increase the recruitment of co-blockers, and suppress the transcription of target genes, ultimately preventing differentiation and leading to the development of APL [206,207]. As the potent transcriptional repressors of retinoic acid signaling, RARα fusion proteins can be overcome with therapeutic doses of ATO [208]. ATO is a common chemotherapeutic agent used primarily for the treatment of refractory or relapsed APL (Figure 13). Although the mechanism of action of ATO is not fully understood, it has been shown to damage or degrade PML–RARα fusion protein and induce apoptosis in MCF-7 cancer cells [209]. Studies have revealed that combining ATRA with chemotherapy has led to APL becoming a highly curable disease with high response rates (>90%) [210]. Furthermore, ATO has been shown to be effective in patients who have relapsed after clinical remission with ATRA [211,212,213] (Figure 13). However, patients carrying the ZBTB16–RARα or STAT5B–RARα fusion gene were naturally resistant to ATO [214,215]. Currently, there are few drugs targeting RARA fusion proteins, and drug resistance remains a significant challenge in treating APL. Developing new therapeutic tools or drugs is necessary to overcome this challenge.

### 4.2. DDIT3-Fusion Inhibitors

DNA damage inducible transcript 3 (DDIT3), a cellular stress sensor, is expressed at low levels under normal physiological conditions. However, it can be rapidly induced in response to various endoplasmic stresses, including reticulum stress, nutrient deprivation, DNA damage, cell growth arrest, or hypoxia. Additionally, DDIT3 is believed to be involved in the negative regulation of cell differentiation [216]. MLS is a prevalent form of liposarcomas distinguished by a characteristic cytogenetic feature of chromosomal translocation. The most frequent gene fusion observed in MLS is FUS–DDIT3, found in over 90% of cases, while EWSR1–DDIT3 is less commonly identified [217,218,219] (Figure 14). The transcriptional regulation of FUS–DDIT3 is controlled by the FUS promoter, whereas mRNA stability relies on the DDIT3 protein. The expression of the FUS–DDIT3 protein is critical for MLS tumor development, and its expression level correlates with the expression of cell proliferation-related genes, although the underlying mechanism remains unknown [217,220]. It was suggested that FUS–DDIT3 could be a potential target for MLS therapy [220]. Furthermore, FUS–DDIT3 activates JAK–STAT signaling, leading to the development of targeted therapies for MLS based on this discovery [218]. Most JAK and GSK-3 inhibitors, such as Fedratinib, TG101209, and AT9283, have been found to be effective [220]. In addition, FUS–DDIT3 induces activation of IGF-IR and PI3K/AKT signaling, and therapies targeting IGF-IR have shown efficacy in treating MLS both in vitro and in vivo. NVP-AEW541, BMS-754807, and Picropodophyllin have demonstrated effective inhibition of IGF-IR and PI3K/AKT signaling activity. These findings suggest that targeting this pathway could be a promising and specific therapeutic approach for the treatment of MLS [221]. Currently, the treatment landscape for MLS has expanded to include the utilization of new agents, such as Trabectedin or Eribulin, in combination with traditional therapeutic approaches [219]. However, a notable deficiency persists in terms of pharmacological interventions that specifically target the FUS–DDIT3 fusion protein, which represents a characteristic molecular alteration in MLS. There is an urgent requirement to develop novel therapeutic strategies that directly address the oncogenic potential conferred by the FUS–DDIT3 fusion protein.

### 4.3. ETS-Fusion Inhibitors

The ETS protein family encompasses 28 transcription factors that possess highly conserved DNA-binding ETS domains. These factors, including FLI1, ERG, FEV, ETV1, ETV4 and others, play a crucial role in regulating a range of physiological processes, including cell proliferation, differentiation, and apoptosis [222]. The role of ETS gene fusions has been widely investigated in prostate cancer, breast cancer, lung cancer, and colorectal cancer [222]. The most common form of ETS gene fusion observed in prostate cancer is TMPRSS2–ERG, accounting for approximately 50% of cases [223]. Moreover, other gene fusions, such as EWS–FLI1, TMPRSG–ERG [224], HNRNPH1–ERG [225], and EWSR1–ETV1 [226] have been observed in different types of cancers (Figure 14). The transcription of ETS is dependent on the expression and activation of poly ADP-ribose polymerase (PARP) and DNA protein kinase (DNAPK). This indicates that as pharmacological agents, PARP and DNAPK shows promising potential in the inhibition of ETS fusion activity [223]. For instance, the PARP inhibitors, such as Olaparib [223], Veliparib, and Rucaparib, have shown therapeutic efficacy against ETS fusions. Rucaparib has been found to be effective in cells with defects in the PTEN gene and expressing ETS gene fusion [227]. YK-4-279 inhibits the activity of EWS–FLI1 fusion protein by blocking RNA Helicase A (RHA) binding with EWS–FLI1 [228]. TK216, a derivative of Yk-4-279, also inhibits the activation of EWS–FLI1 [229]. Dutasteride inhibits the growth of TMPRSS2–ERG fusion-positive cells and significantly slows tumor growth when combined with Bicalutamide [230]. Additionally, the BET protein inhibitor, ABBV- 075, is a potent inhibitor of TMPRSS2–ETS fusion [231]. Studies have shown that the glucocorticoid receptor GR interacts with transcription factors of the ETS family. In addition, the GR antagonist Mifepristone inhibits EWS–FLI1 and EWS–ERG [232]. The diversity of therapeutic approaches for ETS fusions provides valuable insights for follow-up studies.

### 4.4. MYB-Fusion Inhibitors

Avian myeloblastosis viral oncogene homolog (MYB) is a transcriptional regulator that plays a critical role in maintaining normal tissue homeostasis and development [233]. The presence of rearrangements and mutations in the MYB gene serves as compelling evidence for the involvement of MYB family members in human cancers. The most frequently occurring MYB fusion is MYB–NFIB [233,234] (Figure 14). In particular, the AKT-dependent IGF-IR signaling pathway regulates MYB–NFIB, making the IGF-1R–MYB–NFIB axis a promising target for the treatment of ACC [235]. The administration of IGF-1R inhibitors, such as Linsitinib or BMS-754807, has been shown to downregulate the expression of the MYB–NFIB fusion protein [234]. MYB–TYK2 fusion proteins are known to activate the JAK/STAT signaling pathway. In cells expressing MYB–TYK2, a combination therapy involving Vorinostat, Tanespimycin, and Cerdulatinib has exhibited notable efficacy [236]. In addition, transcriptional targets of MYB-QKI, such as KIT or CDK6, can be targeted for therapeutic intervention. ATR, a DNA damage-sensing kinase, has also been reported as a downstream therapeutic target for MYB and MYB–NFIB [234,237]. Monensin can induce MYB degradation and inhibition of MYB–NFIB at both the mRNA and protein levels [234].

### 4.5. MLL-Fusion Inhibitors

Myeloid/lymphoid leukemia (MLL) is a kind of histone methyltransferases in the maintenance of hematopoietic stem cells. MLL gene fusions can be generated due to chromosomal translocation or epigenetic changes, which promote the occurrence and development of leukemia. MLL rearrangement is one of the most prevalent chromosomal abnormalities in AML [238]. Multiple types of MLL fusions were identified, such as MLL–AF4 [239], MLL–AF9 [240,241], MLL–AF6 [242], and others (Figure 14). The oncogenic activity of MLL fusion proteins depends on the binding of the CXXC domain to the promoter and/or enhancer of the target gene. Disulfiram has been shown to disrupt the CXXC domain and inhibit the binding of MLL fusion protein to DNA, thereby attenuating their activity [243]. In addition, most MLL fusion proteins rely on transcriptional regulatory mechanisms mediated by asparagine endopeptidase (AEP) and histone methyltransferase (DOT1L) [241]. A selective DOT1L inhibitor, EPZ004777, has demonstrated efficacy against various MLL fusion proteins, including MLL–AEP [244], MLL–AF9, MLL–AF4, and MLL–ENL fusions [245,246]. Rabeprazole, a proton pump inhibitor, selectively inhibits the growth of MLL–AF4 and MLL–AF9-positive cells [247]. KO-382, a small molecule inhibitor of Menin–MLL interaction, binds to Menin and inhibits the proliferation of leukemic cells transformed by MLL–AF9 [248]. Mocetinostat, the inhibitor of the histone deacetylase complex (HDAC), is used to inhibit MLL–AF9 [246]. The immunoproteasome inhibitor ONX-0914 exhibits therapeutic activity against ALL expressing MLL–AF4 fusion protein [249]. The HSP90 inhibitors Radicol and Ganetespib induce degradation of the MLL fusion protein [250]. In conclusion, the treatment of MLL fusions holds great promise.

### 4.6. FOXO1-Fusion Inhibitors

The forkhead box O (FOXO) family, comprising FOXO1, FOXO3, FOXO4, and FOXO6, represent a group of transcription factors that play critical roles in various physiological processes and responses to environmental stimuli [251]. Among the FOXO family members, FOXO1 fusions are the most prevalent, with notable example including PAX3–FOXO1 [252] and PAX7–FOXO1 [252] (Figure 14). Inhibition of key target genes regulated by PAX3–FOXO1 using the small molecule inhibitor JQ1, which blocks the BRD4 protein, have been shown to induce cell death and effectively suppresses tumor growth in vivo [251]. The potent and selective histone deacetylase inhibitor, Entinostat, reduces the mRNA and protein expression levels of PAX3–FOXO1 in tumor cells. JARID2, a direct transcriptional target of PAX3–FOXO1 fusion protein and its downstream effector, shows reduced cell proliferation upon inhibition [253]. Chromodomain decapping enzyme DNA-binding protein 4 (CHD4) is a key co-regulator of PAX3–FOXO1, meaning it a potential therapeutic target for rhabdomyosarcoma (RMS) [254]. ATR inhibitors show greater activity in PAX7–FOXO1 fusions. Furthermore, PAX3–FOXO1 protein levels can be regulated by proteasome inhibitors such as MG-132. Small molecule inhibitors against fusion transcription factors remain a challenging area of research. Therefore, indirect inhibition strategies against such carcinogens hold promising as a more pragmatic approach.

### 4.7. YAP1-Fusion Inhibitors

Yes-associated protein 1 (YAP1) is an essential transcriptional co-activator that acts as a downstream effector of the HIPPO pathway primarily through the TEAD family of transcription factors [255,256]. It has crucial roles in normal tissue homeostasis, development, and stem cell maintenance, making it an attractive target for cancer therapy [256]. YAP1 fusion types, include YAP1–TFE3 [256], YAP1–MAML2 [256], and YAP1–NUTM1 [255] have been identified (Figure 14). The activity of the YAP1 fusion protein is reliant on its interaction with TEAD, mediated through TEAD binding sites, which has been shown to hinder the development of associated tumors [257]. Therefore, disrupting the interaction between YAP1 and TEAD may hold promise as an effective strategy for cancer therapy. In addition, Verteporfin, a small molecule inhibitor, disrupts the interaction between YAP1 and TEAD, leading to the inhibition of YAP1 function [256]. Pharmacological and genetic disruption of YAP–TEAD interactions reduced the oncogenic potential of the fusion, suggesting the potential of targeting this interaction as a future approach for cancer treatment [258,259].

## 5. Conclusions

In recent years, advancements in detection methods have yielded an expanding repertoire of identified gene fusion events, encompassing both known and novel types. These discoveries have provided valuable insights into potential therapeutic targets, fueling further investigations in this field. Although the efficiency of the assay technology has improved, the challenge of false-positive results persists [30,31]. Hence, continuous refinement of detection methods is critical in deepening our understanding of gene fusions and to foster the development of effective strategies for tumor treatment. Tumors harboring gene fusions usually exhibit heightened malignancy. Despite the existence of numerous FDA-approved inhibitors or therapeutics, the number of drugs available for targeting fusion protein targets, such as EGFR and BRAF fusions, remains limited. BCR–ABL fusion, being one of the earliest discovered fusion types, has relatively mature for development for optimal inhibitor design. Therefore, for other fusion types, it is important to conduct in-depth investigations into the disease mechanisms caused by specific fusions and the factors influencing drug resistance mutations. This knowledge will facilitate the modification or redevelopment of existing compounds. We have compiled a comprehensive list of drugs currently undergoing clinical trials with the specific objective of targeting fusion proteins (Table 1 and Figure 15). Despite the availability of several small molecule inhibitors approved by the FDA, drug resistance remains a pervasive clinical issue [260]. In Table 2, we have provided a summary of the resistance mutation site profiles associated with the existing inhibitors (Table 2). While small molecule inhibitors have undeniably advanced cancer treatment, the exploration and development of alternative therapeutic options are imperative, given the eventual emergence of tumor cells resistance over time [261,262,263].

Protein targeting chimeras (PROTACs) represent a novel technique that effectively induces protein degradation through the ubiquitin–proteasome degradation system (UPS). PROTACs offer potential advantages in addressing target protein mutations, overexpression, and resistance to small molecule inhibitors or monoclonal antibodies [264,265,266]. Moreover, PROTACs hold promise in targeting non-drug or drug-resistant targets, including nuclear receptors, transcription factors, and folded proteins [264,266]. These attributes make PROTACs an attractive approach for treating fusion proteins associated with diseases such as NPM/EML4–ALK and BCR–ABL [266,267], presenting a viable strategy for combating drug resistance. Furthermore, considering that fusion partners often possess coiled-coil domains, crucial for the function of resulting fusion proteins (Figure 4), these domains could serve as potential targets for developing inhibitors in future designs. Moreover, gene fusions have demonstrated remarkable potential in generating tumor-specific neoantigens, making immunotherapy centered around these neoantigens a promising approach for cancer treatment [268]. In particular, certain fusions can modulate the expression of PD-L1 through downstream pathways, as observed with ALK fusions. Given the correlation between PD-L1 expression on tumor cell surfaces and the response to PD-1/PD-L1 antibody, it may be a prudent choice for tumors harboring such gene fusions [153,269]. Among these therapeutic strategies, immune checkpoint blockade (ICB) therapy utilizing PD-1/PD-L1 blockade is currently one of the most prevalent immunotherapeutic approaches. Overall, there is growing optimism that cancers driven by gene fusion can be effectively addressed through continued research and the development of new treatments in the future [270].

## Figures and Tables

**Figure 1 molecules-28-04672-f001:**
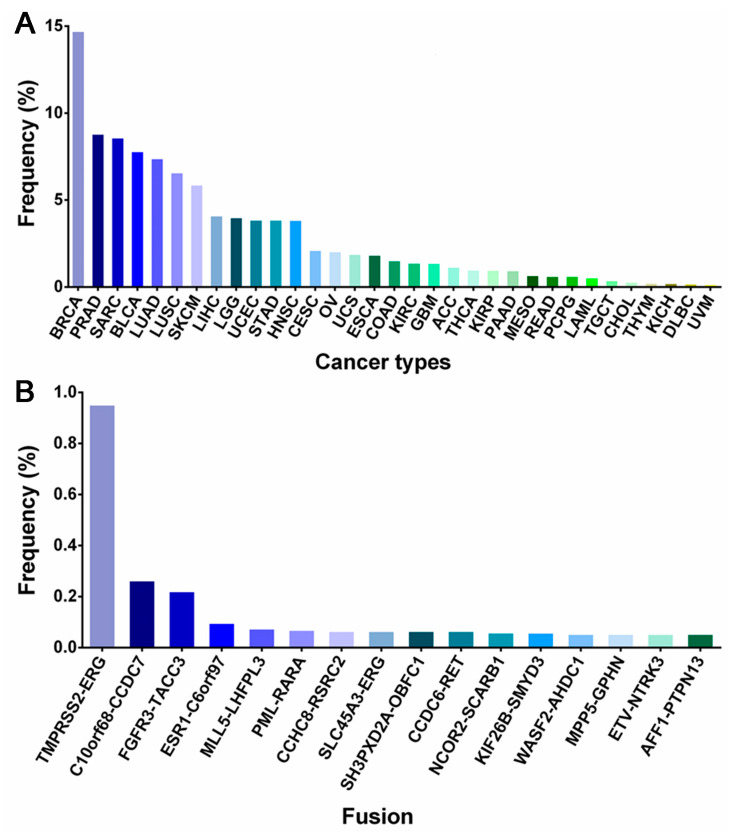
The data in the bar chart show the percentage of gene fusions by cancer types (**A**) and gene fusion types (**B**). The data come from the TCGA Fusion Gene Database [20]. BRCA (Breast invasive carcinoma); PRAD (Prostate adenocarcinoma); SARC (Sarcoma); BLCA (Bladder urothelial carcinoma); LUAD (Lung adenocarcinoma); LUSC (Lung squamous cell carcinoma); SKCM (Skin cutaneous melanoma); LIHC (Liver hepatocellular carcinoma); LGG (Brain lower-grade Glioma); UCEC (Uterine Corpus Endometrial Carcinoma); STAD (Stomach adenocarcinoma); HNSC (Head and Neck squamous cell carcinoma); CESC (Cervical squamous cell carcinoma and endocervical adenocarcinoma); OV (Ovarian serous cystadenocarcinoma); UCS (Uterine carcinosarcoma); ESCA (Esophageal carcinoma); COAD (Colon adenocarcinoma); KIRC (Kidney renal clear cell carcinoma); GBM (Glioblastoma multiforme); ACC (Adrenocortical carcinoma);THCA (Thyroid carcinoma); KIRP (Kidney renal papillary cell carcinoma); PAAD (Pancreatic adenocarcinoma); MESO (Mesothelioma); READ (Rectum adenocarcinoma); PCPG (Pheochromocytoma and Paraganglioma); AML (Acute myeloid leukemia); TGCT (Testicular germ cell tumors); CHOL (Cholangiocarcinoma); THYM (Thymoma); KICH (Kidney chromophobe); DLBC (Lymphoid neoplasm diffuse large B-cell lymphoma); UVM (Uveal Melanoma).

**Figure 2 molecules-28-04672-f002:**
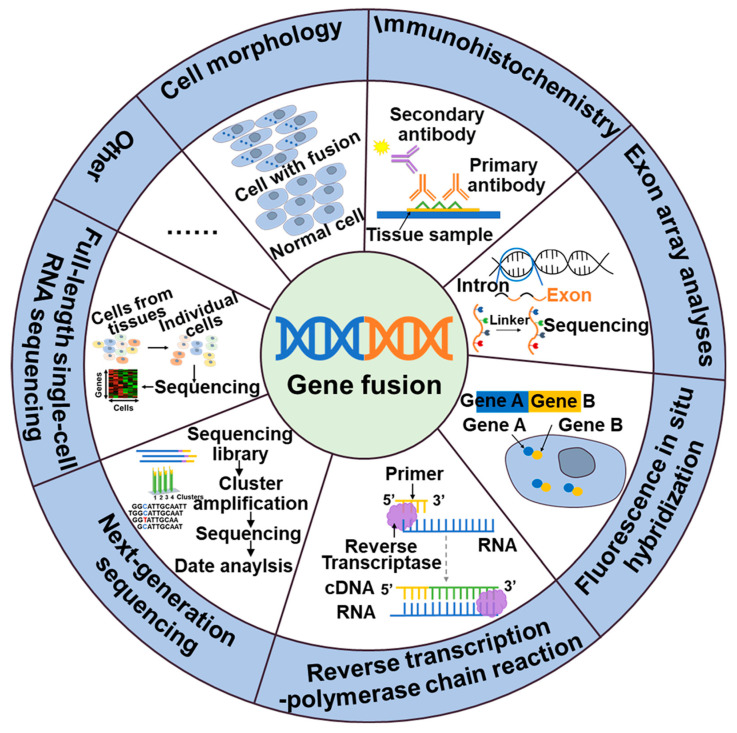
Methods of detection of gene fusions.

**Figure 3 molecules-28-04672-f003:**
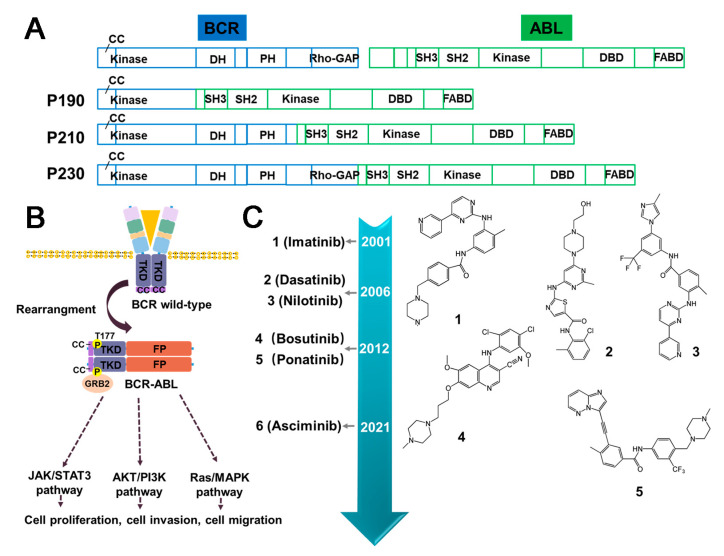
The structure of BCR–ABL fusion protein, its impact on downstream signaling pathways, and the current inhibitors used to target BCR–ABL. (**A**) Domain schematic structural composition of BCR–ABL fusion proteins. (**B**) The activation of molecular pathways downstream of BCR–ABL. Upon dimerization, BCR–ABL fusion proteins undergo autophosphorylation at tyrosine 177 of BCR. The dimerization event facilitates the recruitment of various proteins, such as GRB2, to activate multiple downstream signaling pathways, including PI3K/AKT, MAPK, and JAK/STAT pathways. (**C**) The timeline illustrating the approval of small molecule inhibitors targeting BCR–ABL. CC (Coiled-coil); DH (Dbl-homology); PH (Pleckstrin-homology); SH3/SH2 (Src-homology 3/2); DBD (DNA-Binding domain); RhoGAP (Rho GTPase activating protein); FABD (F-actin binding domain); FP (Fusion partner); TRD (Intracellular kinase domain); P (Phosphorylation sites).

**Figure 4 molecules-28-04672-f004:**
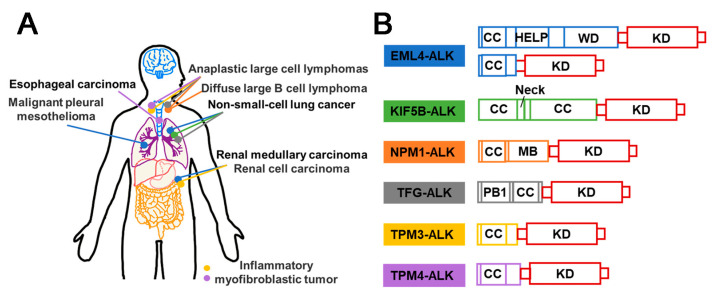
Schematic diagram of ALK fusion proteins in different cancers and their structural composition. (**A**) Illustration of ALK fusion events frequently observed in various cancers. Each colored dot represents a distinct ROS1 fusion event, as depicted in (**B**). (**B**) Domain schematic structural composition of ALK fusion proteins. All the partners in the AKL fusion proteins possess the coiled-coil (CC) domain. HELP (A hydrophobic echinoderm microtubule-associated protein-like protein); MB (A metal binding); PB1 (Phox and Bem 1); KD (Kinase domain).

**Figure 5 molecules-28-04672-f005:**
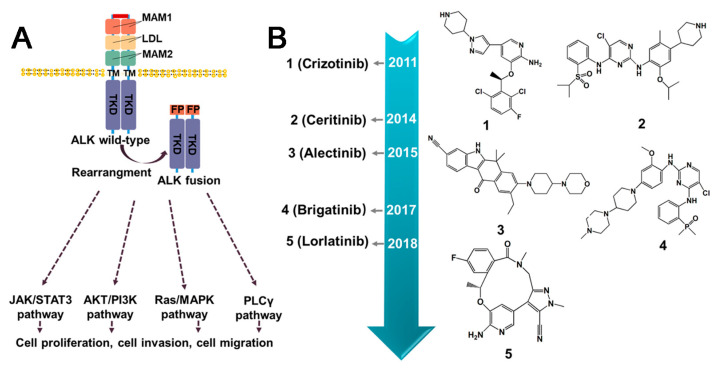
The downstream signaling pathways of ALK fusion proteins and current inhibitors. (**A**) Activation of molecular pathway downstream of ALK fusion proteins. Unlike wild-type ALK, ALK fusion facilitates the ligand-independent dimerization of ALK, resulting in its constitutive activation and the initiation of multiple downstream signaling pathways. (**B**) The timeline illustrating the approval of small molecule inhibitors targeting ALK fusion protein. MAM1/2 (Multiple receptor protein-tyrosine phosphatase mu 1/2); LDL (Low-density lipoprotein receptor domain); TM (Transmembrane domain); FP (Fusion partner); TRD (Intracellular kinase domain).

**Figure 6 molecules-28-04672-f006:**
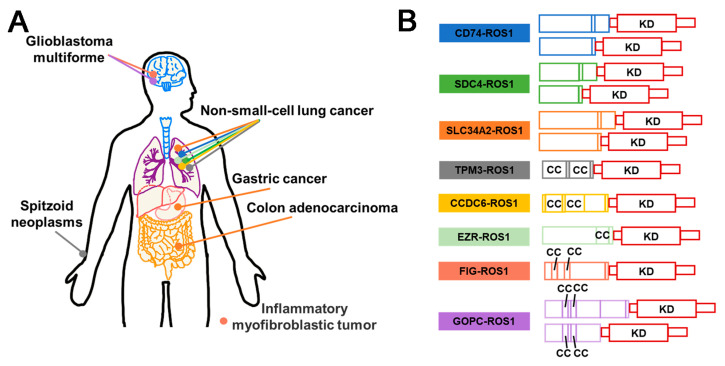
Schematic diagram of ROS1 fusion proteins in cancers and their structural composition. (**A**) Illustration of the frequent occurrence of ROS1 fusion in different cancers. Each colored dot represents a distinct ROS1 fusion event, as depicted in (**B**). (**B**) Domain schematic structural composition of ROS1 fusion proteins. CC (Coiled coil); KD (Kinase domain).

**Figure 7 molecules-28-04672-f007:**
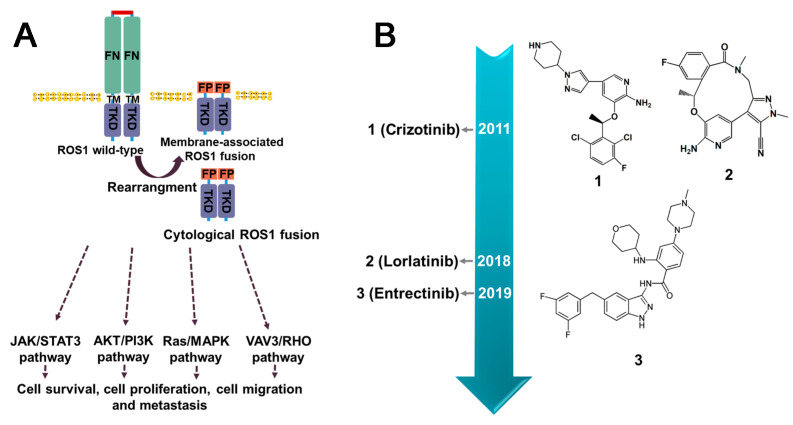
The downstream signaling pathways of ROS1 fusion proteins and current inhibitors. (**A**) The activation of molecular pathways downstream of ROS1 fusion proteins encompasses both ligand-independent cytoplasmic fusion proteins and membrane-associated fusion proteins. (**B**) The timeline for the approval of small molecule inhibitors targeted ROS1 fusion protein. FN (Fibronectin type-III domain); TM (Transmembrane domain); FP (Fusion partner); TRD (Intracellular kinase domain).

**Figure 8 molecules-28-04672-f008:**
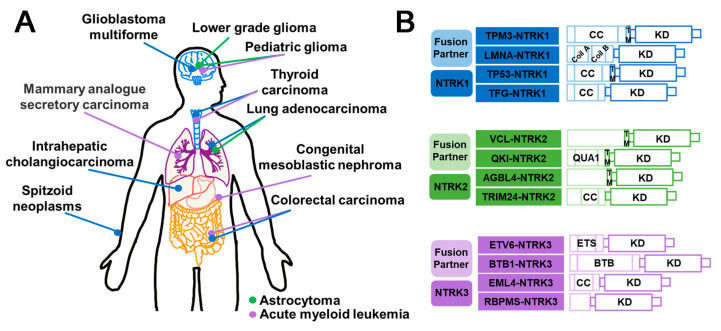
Schematic diagram of NTRK fusions in cancers and their structural composition. (**A**) Illustration of the prevalent NTRK fusion events observed in different cancers. Each distinct colored dot corresponds to a specific NTRK fusion, as depicted in (**B**). (**B**) Domain schematic structural composition of NTRK fusion. CC (Coiled coil); KD (Kinase domain); TM (Transmembrane domain); Coil A/B (Coiled-coil segments A/B); ETS (E twenty-six); BTB (Broad-complex, tramtrack, and bric a brac).

**Figure 9 molecules-28-04672-f009:**
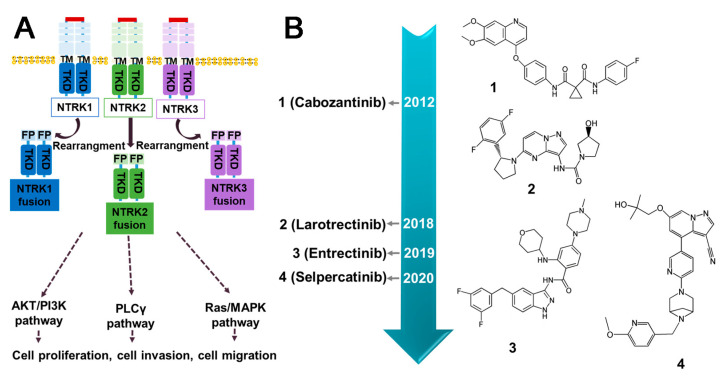
The downstream signaling pathways of NTRK fusions and current inhibitors. (**A**) Activation of molecular pathways downstream of NTRK fusion. All three NTRK proteins have the capability of forming fusion proteins, leading to the activation of their respective signaling pathways. (**B**) The timeline for the approval of small molecule inhibitors targeted NTRK fusion protein. TM (Transmembrane domain); FP (Fusion partner); TRD (Intracellular kinase domain).

**Figure 10 molecules-28-04672-f010:**
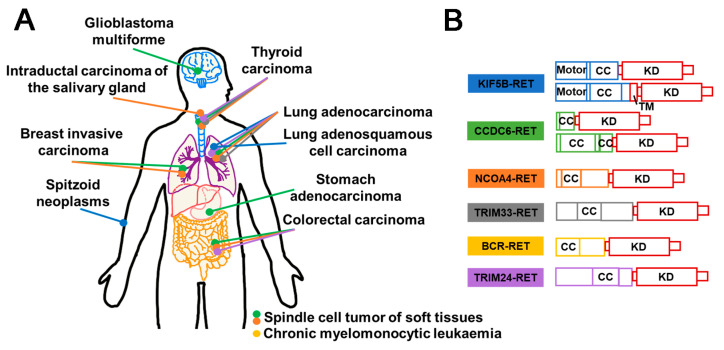
Schematic diagram of RET fusion proteins in different cancers and their structural composition. (**A**) Illustration of the RET fusion events identified in different cancers. Each distinct colored dot corresponds to a specific RET fusion in (**B**). (**B**) Domain schematic structural composition of RET fusion proteins. CC (Coiled coil); KD (Kinase domain); TM (Transmembrane domain).

**Figure 11 molecules-28-04672-f011:**
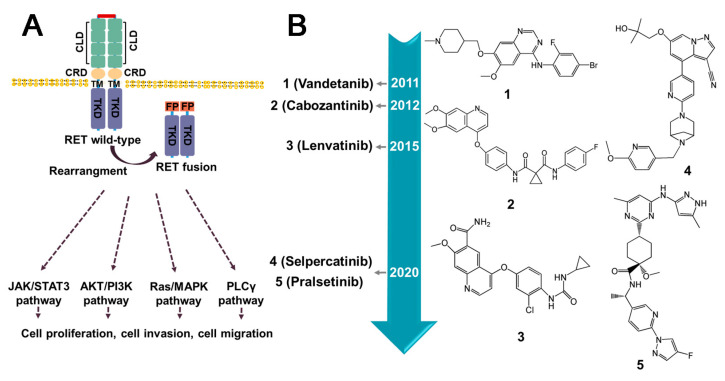
The downstream signaling pathways of RET fusion proteins and current inhibitors. (**A**) Activation of molecular pathway downstream of RET fusion proteins. Most RET fusion proteins facilitate the ligand-independent dimerization of RET kinase domain, resulting in its constitutive activation. (**B**) The timeline for the approval of small molecule inhibitors targeted RET fusion protein. CLD (Cadherin-like domains); CRD (Cysteine-rich domain); TM (Transmembrane domain); FP (Fusion partner); TRD (Intracellular kinase domain).

**Figure 12 molecules-28-04672-f012:**
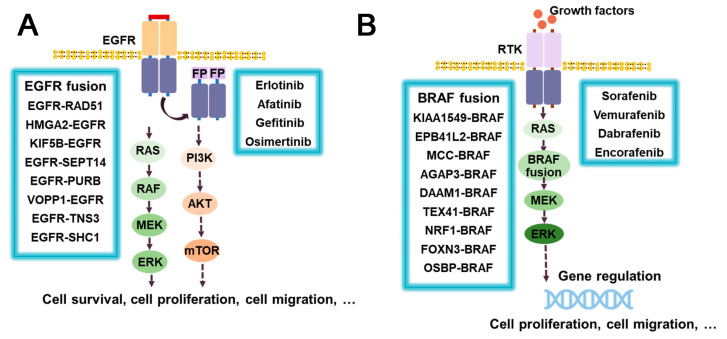
The downstream signaling pathways of EGFR and BRAF fusion proteins, current inhibitors, and major drug resistance mutations: (**A**) EGFR fusion proteins; (**B**) BRAF fusion proteins. RTK (Receptor tyrosine kinase); FP (Fusion partner). The upper right corner of Osimertinib refers to its main drug resistant mutations.

**Figure 13 molecules-28-04672-f013:**
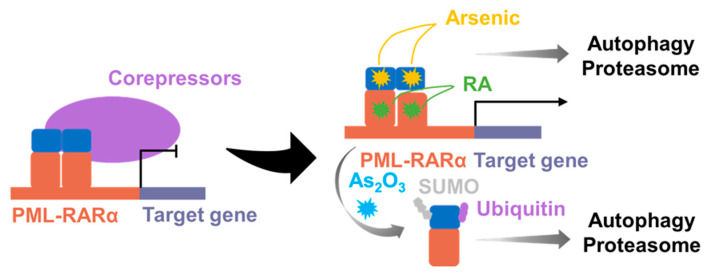
Distinct pathways regulating PML–RARα degradation. Both the ubiquitin–proteasome system and autophagy-mediated degradation can target PML–RARα for degradation. RA: Retinoic acid.

**Figure 14 molecules-28-04672-f014:**
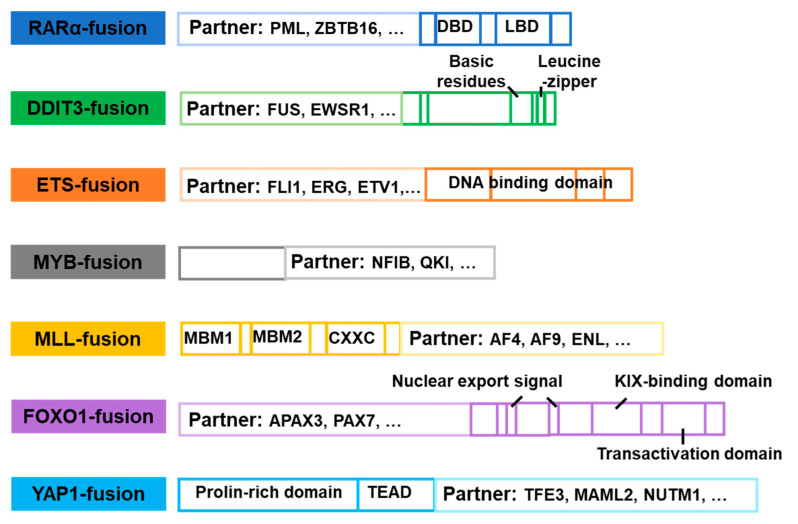
Domain structural composition of common non-kinase fusion genes.

**Figure 15 molecules-28-04672-f015:**
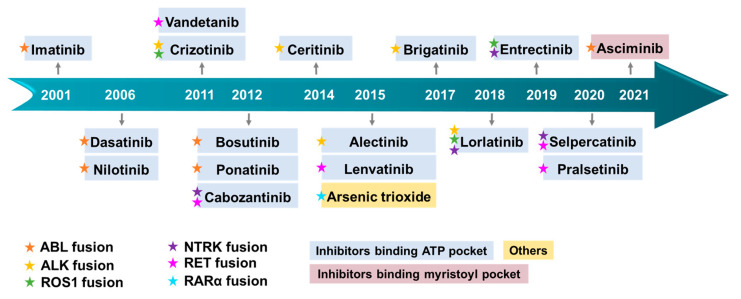
Timeline illustrating the approval of small molecule inhibitors targeting fusion protein. The color of the pentagram symbolizes distinct fusion proteins. Diverse hues of drug bases represent varying mechanisms of action.

**Table 1 molecules-28-04672-t001:** Latest clinical trials of fusion protein inhibitors, treatment modalities, and side effects.

Fusion	Cancer	Drug	Latest Clinical Trials Schedule	Route (Dosage Volume)
ABL fusion (Mainly BCR–ABL)	CML, ALL	Imatinib ***	2021	Oral (Tablet, film coated)
		Dasatinib ***	2021	Oral (Tablet, film coated)
		Nilotinib ***	2021	Oral (Capsule)
		Bosutinib ***	2021	Oral (Tablet, film coated)
		Ponatinib **	2021	Oral (Tablet, film coated)
		Asciminib **	2021	Oral (Tablet, film coated)
ALK fusion	NSCLC, IMT	Crizotinib **	2014	Oral (Capsule)
		Ceritinib **	2015	Oral (Tablet, film coated; Capsule)
		Alectinib **	2021	Oral (Capsule)
		Brigatinib **	2023 ^&^	Oral (Tablet, film coated)
		Lorlatinib **	2023 ^&&^	Oral (Tablet, film coated)
ROS1 fusion	NSCLC, Other solid tumors	Crizotinib ***	2019	Oral (Capsule)
		Entrectinib **	2020	Oral (Capsule)
		Lorlatinib *	2023	Oral (Tablet, film coated)
NTRK1/2/3 fusion	GIST, NSCLC, IFS	Entrectinib *	2021	Oral (Capsule)
		Larotrectinib **	2021	Oral (Capsule; Solution, concentrate)
		Cabozantinib **	2022	Oral (Tablet, film coated)
		Selpercatinib **	2023 ^&&^	Oral (Capsule)
RET fusion	NSCLC, THCA, HCC, MTC	Lenvatinib **	2013	Oral (Capsule)
		Cabozantinib **	2012	Oral (Tablet, film coated; Capsule)
		Vandetanib **	2013	Oral (Tablet, filmcoated)
		Selpercatinib **	2020	Oral (Capsule)
		Pralsetinib **	2021	Oral (Capsule)
EGFR fusion	NSCLC, PAAD	Gefitinib	/	Oral( Tablet, film coated)
		Erlotinib	/	Oral (Tablet, film coated)
		Afatinib	/	Oral (Tablet, film coated)
		Osimertinib	/	Oral (Tablet, film coated)
BRAF fusion	NSCLC, CRC, HCC, GIST	Sorafenib	/	Oral (Tablet, film coated)
		Vemurafenib	/	Oral (Tablet, film coated)
		Dabrafenib	/	Oral (Tablet; for suspension, Capsule)
		Encorafenib	/	Oral (Capsule)
RARα fusion	APL	Arsenic trioxide ***	2015	Inject (Injection)

*: Phase I clinical; **: Phase II/III clinical; ***: Phase IV clinical; &: Not yet recruiting; &&: Recruiting. Drugs without superscripts indicate activity against this type of fusion but no clinical trial data are available at this time. GIST (Gastrointestinal mesenchymal tumor); IFS (Infantile fibrosarcoma). Data source: www.fda.gov, go.drugbank.com and clinicaltrials.gov (Cutoff date: 29 May 2023).

**Table 2 molecules-28-04672-t002:** Resistance mutation sites for the inhibitors of fusion proteins mentioned in this paper.

Fusion Proteins	Drug	Resistance Mutations
ABL fusion (Mainly BCR–ABL)	Imatinib	T315I, E255K, Y253F, M351T
	Dasatinib	T315I, F317L
	Nilotinib	T315I, G250E
	Asciminib	V468F, I502L
ALK-fusion	Crizotinib	1151Tins, G1202R, S1206Y, L1152R, C1156Y, L1196M, I1171T, F1174V, L1198P, D1203N, F1174L/C, G1269A
	Ceritinib	G1123S, G1202R, F1174C
	Alectinib	I1171T/N/S, G1202R, V1180L
	Brigatinib	G1202R
ROS1 fusion	Crizotinib	S1986Y/F, G2032R, L2155S, F2004C/I/V
	Entrectinib	G2032R, F2004C/I/V, L2086F, L2026M
NTRK1/2/3 fusion	Entrectinib	G667C, G696A, G595R, G623R/E, V573M, G595L
	Larotrectinib	/
RET fusion	Lenvatinib	V738A, A807V, F998V, L730V, E732K, G810R/S/C/V/A
	Cabozantinib	L730I, E732K, V871I
	Vandetanib	G810A/S/R/V, L881V, C634W, M918T, V804L/M/E, S904F
	Selpercatinib	G810R/S/C/V
	Pralsetinib	G810R/S
EGFR fusion	Gefitinib	T790M
	Erlotinib	T790M
	Afatinib	T790M
	Osimertinib	L792, G796, C797
BRAF fusion	Sorafenib	V600E, G12D, G13D
	Vemurafenib	G12D
	Dabrafenib	/
	Encorafenib	/

## Data Availability

Not applicable.

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
