# Peer review of "Small Molecule Inhibitors as Therapeutic Agents Targeting Oncogenic Fusion Proteins: Current Status and Clinical"

_molecules, 2023, doi:10.3390/molecules28124672_

Round 1

Reviewer 1 Report

The review of Kong Y et al., "Small molecule inhibitors as therapeutic agents targeting oncogenic fusion proteins: current status and clinical" represents a comprehensive compendium of current information on the small molecules targeting main products of oncogenic translocations. The review will be interesting for the broad auditory including scientist and clinicians. 

Minor comments:

1. The font and small imagens in Fig. 2, Fig. 3A,c, Fig. 8B, Fig, 13, and Fig. 14 are not readable. Please increase. 

2. Line 183 "...structural domain of BCR-ABL kinase...". Which kinase domain S/T-kinase or Y-kinase?

3. Line 202 "PDF". PDGF?

4. Line 647. Please provide the explanation for the abbreviation ATO.

Reviewer 2 Report

The authors reviewed the development of small molecules targeting kinase and non-kinase fusion genes. The main limitation of the manuscript is the absence of criteria used in the cited articles selection. The authors did not report the period, the keywords and the database used in their search. This information must be available in the introduction of the manuscript. A review article without this information is not reproducible. In addition, the abstract is poorly informative due to the absence of the main scaffolds explored in the development of new oncogenic fusion proteins inhibitors, the number of new chemical entities developed, and the years in which the first molecules were developed and reached the market. In addition to this, other aspects to be considered by the authors are listed below:

(1) The quality of figures must be improved before a possible publication: in figures 1, 2 and 3 is very difficult the reading of y-axis due to the small font size. The chemical structures are very small and is not possible to see the functional groups. The structures are not numbered.

(2) Is suggested the construction of a timeline depicting the years in which each new molecule emerged as potential new drug.

(3) Is missing the analysis of the chemical diversity of the molecules approached in the manuscript. The summarization of this aspect may be used as a guide for the design of new molecules targeting oncogenic fusion.

(4) In a large part of the manuscript, the authors described the development of each drug. The text is exhaustive and the standard was kept in all the manuscript. I suggest making the manuscript more dynamic and improving the integration of the text with the figures. In some cases, abbreviations depicted in the figures were not described or discussed in the text. 

(5) In a large part of the text, may be found cataloging and description of findings without critical reviewing. This is not the kind of review article able to attract readers, so is more productive to make a search on scientific databases than read this kind of article. Please see that recently emerged in the academic environment questions about the quality and the type of reviews articles that are being published. For a detailed discussion, see the opinion article Front. Plant Sci., 26 April 2016, DOI: 10.3389/fpls.2016.00494, title: “Yes, We Have an Inflation of Reviews: But of the Wrong Kind!”

Reviewer 3 Report

Authors discussed the current state of small molecular inhibitors as therapeutic agents for oncogenic fusion proteins, including related and basic mechanisms, and related clinical trials. They also discussed the challenges associated with their utilization. They claimed the objective is to provide the medicinal community with current and pertinent information and to expedite the drug discovery programs in such field. Apparent flaws exist in this manuscript. major revision is recommended before acceptance. Main issues have been listed as following.

1.It is extremely confused that the title of this manuscript was named as “agents targeting oncogenic fusion proteins”, however their main part as section 3 was actually titled as “inhibitors targeted kinase fusion genes”. These two therapeutic strategies are different ï¼ˆhttps://doi.org/10.1016/B978-0-12-820595-2.00011-4)( doi.org/10.1016/j.celrep.2021.109647.) .

2.A figure that summarizes all main types of selected inhibitors should be added. 

3.Main informative and conclusive content should be added for each figure in figure legends.

4.Started date (or finished date) of selected clinical trials should be added in table 1. Moreover, the main outcomes of finished trials should also be added in this table, and be discussed in manuscript. 

5.Please also summarize the common side effects of fusion proteins inhibitors based on table1, and provide therapeutic approaches.

English should be refined by professional and native speaker.

Round 2

Reviewer 2 Report

I am satisfied with the answers of the author. Thus, in my opinion, the manuscript meets the requirements for publication in Molecules, and I recommend accept it in its current form. 

Minor editing of English language required

Reviewer 3 Report

Authors discussed the current state of small molecular inhibitors as therapeutic agents for oncogenic fusion proteins, including related and basic mechanisms, and related clinical trials. They also discussed the challenges associated with their utilization. They claimed the objective is to provide the medicinal community with current and pertinent information and to expedite the drug discovery programs in such field. 

Minor editing of English language is recommended.